



# Short-term effects of hurricanes on nitrate-nitrogen runoff loading: a case study of Hurricane Ida using E3SM land model (v2.1)

Yilin Fang[1], Hoang Viet Tran[2], and L. Ruby Leung[2]

[1]Earth System and Science Division, Pacific Northwest National Laboratory, Richland, WA, USA
[2]Atmospheric Sciences and Global Change Division, Pacific Northwest National Laboratory, Richland, WA, USA

*Correspondence to*: Yilin Fang (yilin.fang@pnnl.gov)

**Abstract.** When nutrient level in the soil surpasses vegetation demand, nutrient losses due to surface runoff and subsurface leaching are the major reasons for the deterioration of water quality. The Lower Mississippi river basin (LMRB) is one of the sub-basins that deliver the highest nitrogen loads to the Gulf of Mexico. Potential changes in episodic events induced by

hurricanes may exacerbate water quality issue in the future. However, uncertainties in modeling the hydrologic response to hurricanes may limit the modeling of nutrient losses during such events. Using a machine learning approach, we calibrated the land component of the Energy Exascale Earth System model (E3SM), or ELM, version 2.1, based on the water table depth (WTD) of a calibrated 3D subsurface hydrology model. While the overall performance of the

calibrated ELM is satisfactory, some discrepancies in WTD remain in slope areas with low precipitation due to the missing lateral flow process in ELM. Simulations including biogeochemistry performed using ELM with and without model calibration showed important influences of soil hydrology, precipitation intensity, and runoff parameterization on the magnitude of nitrogen runoff loss and leaching pathway. Despite such sensitivies, both ELM

simulations produced reduced WTD and increased runoff and accelerated nitrate-nitrogen runoff loading during Hurricane Ida in August 2021, consistent with the observations. With observations suggesting more pronounced effects of Hurricane Ida on nitrogen runoff than the simulations, we identified factors for model improvement to provide a useful tool for studying hurricane-induced nutrient losses in the LMRB region.





## 1. Introduction

Tropical cyclones are projected to be more intense and potentially make more frequent landfall in some coastal regions in the future due to global warming (Knutson et al., 2020; Pérez-Alarcón et al., 2023; Balaguru *et al.*, 2023). Hurricanes can cause wide-spread, acute disturbances for coastal aquatic and terrestrial ecosystems (Valiela *et al.*, 1998). Besides

catastrophic flooding, enhanced nutrient input coupled with increased runoff were often observed as a result of the heavy precipitation associated with landfalling hurricanes in coastal regions. For instance, five days after Hurricane Katrina made landfall in August 2005, the mean bay-wide nitrate concentration increased by 5.2-fold over the pre-hurricane levels in Biscayne Bay, Florida (Zhang *et al.*, 2009). In a forested watershed draining into Chesapeake Bay, Hurricane Irene in

August 2011 caused an increase of total nitrogen on the rising limb of the storm compared to the baseflow levels (Vidon *et al.*, 2018). High discharge due to Hurricane Irene in 2011 also resulted in high nutrients loading to Newark Bay in northern New Jersey (Nie *et al.*, 2023). The loss of vegetation, attributed to Hurricane Hugo, led to a 108-154% increase in exported nutrients primarily due to increased outflow during the hurricane (Wilson et al., 2006). In coastal North

Carolina, nutrient loadings coincided with the increases of freshwater discharge associated with recent tropical storms (Paerl *et al.*, 2020). Additionally, a model simulation suggests that immediate surges of heavy precipitation associated with hurricanes accelerate nitrogen export more than the long-term average (Sun *et al.*, 2022).

Besides the episodic influence of hurricanes in coastal regions, riverine nitrogen (N) loading

from agricultural lands upstream can lead to significant soil fertility depletion and degradation of water quality in downstream aquatic ecosystems (Li *et al.*, 2022). Excessive nutrient loads can contribute to eutrophication, leading to adverse effects on aquatic ecosystems and water quality (Carpenter *et al.*, 1998). For instance, excessive nutrient loading from cropland in the Mississippi River Basin is a significant contributing factor to the formation of the hypoxic zone in the

northern Gulf of Mexico (Ritter and Chitikela, 2020). Assessing the immediate and long-term impact of hurricanes on water quality in the affected ecosystems is challenging due to logistical constraints associated with sampling during these events (Filippino *et al.*, 2017). A thorough understanding of the mechanisms governing nutrient export from agricultural watersheds will be crucial in managing nutrient pollution, especially in light of the expected hydrological

modifications due to a shifting climate (Speir *et al.*, 2021).





Earth system models have the capability to simulate the coupled carbon and nitrogen cycles and river nitrogen (Nevison *et al.*, 2016). However, there remains a research gap regarding the Earth system model's capability to accurately predict the impact of hurricanes on nitrogen river loading, particularly considering the spatial heterogeneity and temporal variability of

precipitation patterns associated with hurricanes. Understanding the driving mechanism behind excessive riverine N loading during hurricanes, i.e., through surface runoff or groundwater flow, is also lacking. Addressing these gaps is critical for improving our understanding of nutrient transport dynamics and enhancing the capabilities of Earth system models in regions affected by storms. This study investigates the short-time effect of hurricanes on nitrogen loading in runoff,

with an emphasis on how such effect is influenced by soil hydrology and its representations in Earth system models. Using Hurricane Ida as an example, we simulate its impact on nitrate-nitrogen runoff loading in the Lower Mississippi River basin (LMRB) using the land model of the Energy Exascale Earth System (E3SM) (Golaz et al., 2019). We will first describe the model and calibration of the runoff parameterizations using a machine learning approach. The model is

used to assess the transient effects of Hurricane Ida on hydrological and nitrogen river loading in the LMRB, which extends into the Gulf of Mexico. Comparison of model simulations with and without calibration provides insights on the sensitivity of the hydrologic response and nutrient losses to soil hydrology and its representations in models to inform future development needs.

## 2. Methods


### 2.1 Study area

The LMRB, with almost 4 million hectares of irrigated cropland spanning six southern U.S. states, plays a crucial role in the economic landscape. The LMRB is characterized by a humid subtropical climate and significant soil and precipitation variations (Reba and Massey, 2020).

For example, the LMRB experiences varying annual average rainfall ranging from approximately 1143 mm in the north to about 1524 mm in the southern coastal region (Nelson et al., 2022). Cropland is the dominant land cover type in the LMRB. Agriculture relies heavily on the Mississippi River Valley alluvial aquifer to provide over 90% of the irrigation water because a majority of precipitation falls during the winter and spring (Reba and Massey, 2020).

Furthermore, within the last 20 years, the LMRB has been subjected to cyclical flooding events





and declines in groundwater levels due to extreme climate events, leading to the degradation of surface water quality during flooding (Ouyang et al., 2020). The strongest hurricanes to hit the LMRB on record is Hurricane Ida, which formed on 26 August 2021 and made landfall on 29 August 2021 (Fig. 1). Ida had a weak post landfall decay rate, retaining hurricane intensity even

12 h after landfall, potentially due to high soil moisture content ahead of Ida that provided a source of atmospheric moisture and latent energy to fuel the storm (Zhu *et al.*, 2022).

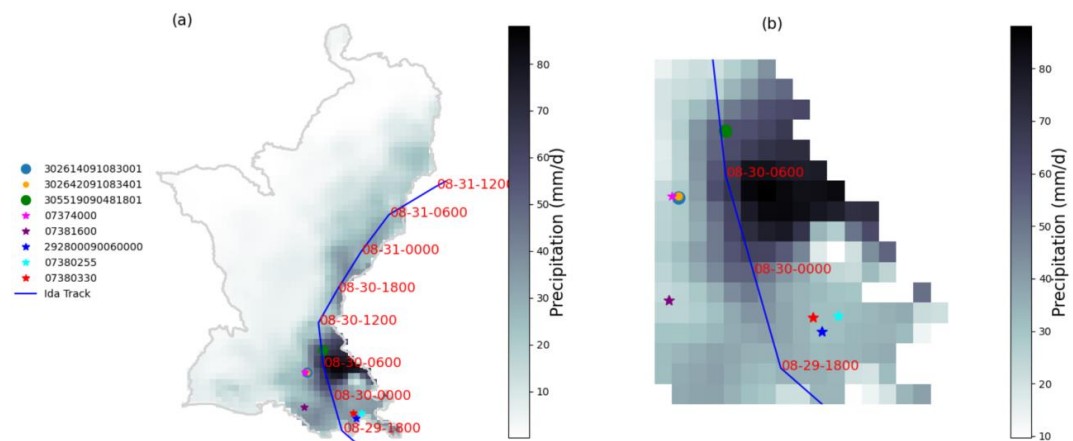

**Figure 1. (a) Average precipitation rate in August 2021 in the Lower Mississippi River Basin and**
**the locations of observation stations along the path of Hurricane Ida (blue line). The red numbers along the path represent the timing of the 6-hourly locations of Ida along its track. Dots are water table stations, and stars are stream water quality stations. Note that stations 302614091083001, 302642091083401, and 07374000 are in close proximity to each other as shown in (b), a zoomed-in view of the southeastern subregion in (a).**

**2.2 Data from measurements and the model simulation**

In the LMRB, hydrologic data (streamflow, groundwater level) are provided by the U.S. Geological Survey's (USGS) National Water Information System (NWIS), while water quality data are obtained from the Water Quality Portal (WQP). WQP currently houses data from the USGS, Environmental Protection Agency (EPA), and U.S. Department of Agriculture (USDA).

Groundwater levels are from monitoring wells including stations 302614091083001 and 302642091083401 in East Baton Rouge Parish, and station 305519090481801 in St. Helena Parish, Louisiana. Water quality monitoring locations include 07374000 associated with a stream in West Baton Rouge Parish, Louisiana, 07381600 associated with a stream in St. Mary Parish,



Louisiana, 292800090060000 associated with an impoundment in Jefferson Parish, Louisiana,

07380255 associated with a stream in Jefferson Parish, Louisiana, and station 07380330
associated with an Estuary in Lafourche Parish, Louisiana. The locations of these monitoring
stations are shown in Figure 1.

We also make use of groundwater level simulated by an integrated surface-subsurface
hydrologic model from our previous effort to investigate the impacts of land cover change on the

hydrologic response to Hurricane Ida in the LMRB (Tran et al., 2024). The integrated surface-
subsurface hydrologic model, ELM-ParFlow, couples the Energy Exascale Earth System Model
(E3SM) land model (ELM) and the three-dimensional subsurface hydrology model ParFlow
(Fang et al., 2022). ParFlow integrates three-dimensional subsurface flow with overland flow
using physics-based equations (Kollet & Maxwell, 2006; Maxwell, 2013; Maxwell & Miller,

2005).  ELM-ParFlow was developed to address the subsurface lateral flow, or the movement of
water through soils and bedrock on hillslopes, which is often missing in Earth system models
that adopt one-dimensional land surface models. The study conducted by Tran et al. (2024)
employed ELM-ParFlow to investigate the relative influence of the changes in surface runoff
versus evapotranspiration due to land cover change on streamflow in inland areas during

hurricane events. Changes in soil hydrology due to land cover change, as examined by Tran et al.
(2024), or due to model representations of soil hydrology, as to be investigated below, can lead
to significant alterations in soil water, with important implications for soil biogeochemistry and
nitrogen river loading.

**2.3 Energy Exascale Earth System Model (E3SM) land model (ELM) (v2.1)**

Derived from the Community Land Model CLM4.5 (Oleson et al., 2013), ELM has been
enhanced with additional features, specifically addressing soil hydrology and biogeochemistry,
as described in Golaz et al., (2019) and Burrows et al., (2020). Operating at the grid-cell level,
ELM delineates the land surface into multiple soil layers and plant functional types. Relevant
hydrological processes in ELM for this study include changes in surface water, canopy water,

soil water, and snow water through interception, throughfall, canopy drip, snow accumulation
and melt, infiltration, evapotranspiration, runoff, redistribution of water within the soil column,
as well as groundwater discharge and recharge. Similar to other global land surface and Earth
system models, soil hydrology in ELM is simulated through 1D columns, with no interaction





between grid cells. The runoff generation in ELM is based on the simple TOPMODEL-based

runoff parameterization (Niu et al., 2005).

The biogeochemical configuration of ELM, or ELM-BGC, is designed to simulate various biogeochemical processes (Burrows et al., 2020). The model simulates active plant phenology and incorporates nutrient controls on vegetation photosynthesis and includes multiple prognostic pools for carbon, nitrogen, and phosphorus within vegetation, litter, and soil organic matter. Two

representations of terrestrial carbon-nitrogen-phosphorus coupling are incorporated into the model: the conceptual Convergent Trophic Cascade (CTC) approach (Yang *et al.*, 2016; Duarte *et al.*, 2017) and the mechanistic Equilibrium Chemistry Approximation (ECA) approach (Tang, 2015; Medvigy *et al.*, 2019). Details of the approaches can be found in Burrows et al. (2020) and the citations therein. The CTC representation is the default option in the model, which is used in

this study.

In ELM-BGC, mineral nitrogen transformations include competition among plant uptake for growth, nitrogen mineralization, microbial immobilization (nitrogen taken up by soil organisms, limited by the availability of mineral nitrogen), denitrification and nitrification (Oleson et al., 2013). Mineral nitrogen that remained in the soil is subject to loss due to leaching from land to

rivers and oceans. The leaching is assumed to act only on nitrate-nitrogen pools. Total nitrogen leaching includes soil nitrogen loss by surface runoff and leaching by subsurface drainage, which is represented by the equation below as a general form:

$$F_N = \frac{QN_{sminn}}{WS_{soil}} \qquad (1)$$

where $F_N$ is the soil nitrogen runoff or leaching, $Q$ is the surface runoff or subsurface drainage,

$N_{sminn}$ is the soil mineral nitrogen, and $WS_{soil}$ is the water storage in soil. The subsurface nitrogen leaching is limited on each time step to not exceed that in the soil.

## 2.4 ELM Calibration

Because groundwater depths can significantly influence soil nutrient concentrations in various ecosystems (Hefting *et al.*, 2004; Miao *et al.*, 2013; Jasinski *et al.*, 2022; Zhang *et al.*,

2022), we first calibrated ELM based on the groundwater table depth (WTD) simulated by ELM-ParFlow (Fang et al., 2022; Tran et al., 2024) in which lateral hydrological flow was simulated



explicitly by the 3D subsurface flow and overland flow. The ELM-ParFlow simulation at 90-m
grid resolution was reported in Tran et al. (2024), hence referred to as Tran2024 hereafter, and
corresponds to the simulation with current land cover which has been evaluated using observed

streamflow data. To capture the ELM grid-level groundwater table dynamics due to both vertical
and lateral hydrological flow processes represented in Tran2024 using ELM-ParFlow, we
calibrated the parameter values of the ELM parameterizations of surface and subsurface runoff
as shown in the equations below.

$$R_{over} = q_{liq}f_{max}e^{(-0.5f_{over}z_{\nabla})} \tag{2}$$

where $R_{over}$ is the surface runoff, $q_{liq}$ is the flux of water reaching the soil surface from the top,

$f_{max}$ is the maximum saturation fraction at a given grid cell. $f_{over}$ is a decay factor, $z_{\nabla}$ is the
groundwater table depth.

$$R_{drain} = q_{drain,max}e^{(-0.5f_{drai}z_{\nabla})} \tag{3}$$

where $R_{drain}$ is the subsurface runoff, $q_{drain,max}$ is the maximum drainage rate, $f_{drai}$ is the decay
factor. For a given $z_{\nabla}$, the larger $f_{drai}$ results in lower $R_{drain}$ and vice versa. By default, $f_{max}$ ranges
from 0.23 to 0.58, and $f_{drai} = 2.5$.

To match the groundwater table depth (WTD) in Tran2024, the total runoff is adjusted by
estimating the maximum saturation fraction, $f_{max}$, and the decay factor, $f_{drai}$, for each ELM grid
cell. The calibration is performed using machine learning, in particular, neural network, where
atmospheric forcing, topography, grid-level average of the WTD of Tran2024 are the predictors,
and $f_{max}$ and $f_{drai}$ are the targets. The initial training dataset of WTD was from an ELM simulation

in which the values of $f_{max}$ and $f_{drai}$ are randomly assigned to each grid, assuming uniform
distributions of $f_{max}$ within the range between 0 to 1 and $f_{drai}$ within the range of 1e-5 to 100. To
estimate $f_{max}$ and $f_{drai}$ given the grid-level temporal average of WTD in Tran2024 at each ELM
grid, the following procedures are taken to iteratively improve the parameter estimation by
updating the training dataset at each iteration:

1) Construct a neural network using the initial training dataset of ELM WTD, atmospheric
forcing, and topography as the predictors to train the model to predict $f_{max}$ and $f_{drai}$ corresponding
to the ELM WTD as the targets. The root mean square error (RMSE) between the predicted and
randomly prescribed values of $f_{max}$ and $f_{drai}$ within the simulation domain serves as the loss
function in this training process.





2) Use the trained neural network model by replacing the ELM WTD with the Tran2024 WTD as the predictors to predict new $f_{max}$ and $f_{drai}$, and use the predicted $f_{max}$ and $f_{drai}$ to update the ELM WTD by running ELM with the new $f_{max}$ and $f_{drai}$.

3) Combine the updated ELM WTD values with the atmospheric forcing and topography as predictors, and the new $f_{max}$ and $f_{drain}$ predicted in step 2) as targets. Merge this combined dataset

with the previous dataset in steps 1) and 2), retrain the neural network model using the updated dataset. Combining the improved data from the previous iterations can increase the model's training dataset size, enable the model to learn and adapt to the more complex patterns to better represent the underlying relationship in the data.

4) Continuously refine the neural network model by repeating steps 2 and 3 iteratively until the

predicted $f_{max}$ and $f_{drai}$ converge to a point where the correlation between ELM WTD and Tran2024 WTD cannot be significantly improved from the previous iteration. When the iterations steps exceed four, only the newest five datasets are included to refine the model. The neural network model includes two hidden layers with 128 neurons each, and an output layer with 2 neurons. ReLU activation function is used to introduce nonlinearity. A dropout layer with

a dropout rate of 25% is inserted to mitigate overfitting by randomly deactivating neurons during training. The Adam optimizer with a learning rate of 1e-5 is used for optimization. 80% of the dataset is used as training data and processed in batches of size 128 over 3000 epochs.

### 2.5 ELM-BGC simulations

The atmospheric forcing, including precipitation, air temperature, shortwave and longwave

radiation, wind speed, specific humidity, and atmospheric pressure, used to drive the ELM simulations is from the North American Land Data Assimilation System (NLDAS) project at 1/8th-degree grid spacing. The resolution of the ELM simulation domain is also set at 1/8th-degree. The ELM hydrology simulation was driven by forcing data spanning from 1980 to 2022. The forcing from year 1980 was used repeatedly for the 600-year spin-up of ELM-BGC for the

default model. Transient simulation of ELM-BGC was then conducted from 1980 to 2020. The result at the end of 2019 was used as initial condition for the model comparison between the default and calibrated ELM. Land cover information was derived from the Land Use Harmonized version 2 (LUH2) (Hurtt et al., 2021).





In this study, we only consider nitrate-nitrogen loading in runoff due to natural terrestrial N
inputs during Hurricane Ida, and crop management (e.g., fertilization and irrigation) are not
considered. The impacted regions from Hurricane Ida will be examined. The selection criteria of
the impacted regions include grids with accumulated precipitation exceeding 17.3 mm d-1 in the
eastern part of the domain from the time of Ida's landfall until the end of August.

## 3. Results


### 3.1 Calibration of ELM parameter values

A satisfactory match between the WTD from the ELM simulation and Tran2024 is achieved
within 10 iterations of parameter estimation using the machine learning and ELM simulations, as
described in Section 2.4. Figure 2 shows a comparison of the $f_{max}$ and $f_{drai}$ used to perform ELM
simulation before the last iteration and those estimated using machine learning during the last
iteration to match the WTD from the ELM simulation with the WTD from Tran2024, with $R^2$
values of 0.87 and 0.93, respectively. The machine learning model has a decent performance
considering the heterogeneity in topography and precipitation (Figs. 3a and 3d) within the
simulated domain.

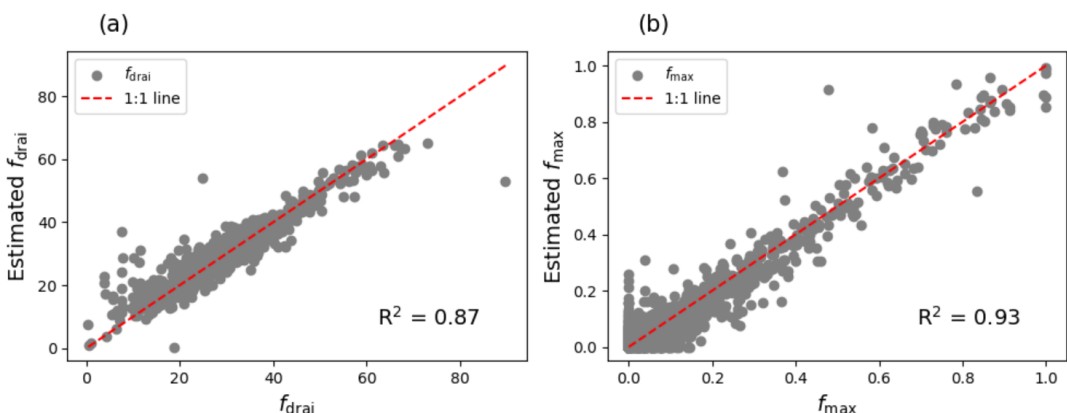


**Figure 2. Scatter plot of estimated and prescribed decay factor, $f_{drai}$ (a) and the maximum
saturation fraction, $f_{max}$ (b) in the last iteration.**

In a majority of the domain, the estimated $f_{max}$ values are nearly 0 (Fig. 3b) and $f_{drai}$ (Fig. 3c)
follows the pattern of the elevation (Fig. 3a). In the midwestern part of the domain, high
elevation and precipitation lead to large $f_{max}$ and low $f_{drai}$, consequently large runoff based on
Eqs. 2 and 3 in those grid cells. In the western slope area with low precipitation (indicated by the



red colored area in Fig. 3d), ELM WTD is deeper than in Tran2024 (Fig. 3e). This occurs even when $f_{max}$ is approximately 0 and $f_{drai}$ is high, favoring nearly no runoff. This result suggests that these areas receive water from wetter areas at high elevations through lateral flow represented by

ELM-ParFlow in Tran2024, which cannot be represented in the 1D ELM through simple calibration of the parameters related to the runoff parameterizations. The overpredictions of WTD in other areas (Figs. 3e and 3f) are primarily due to the same reason.

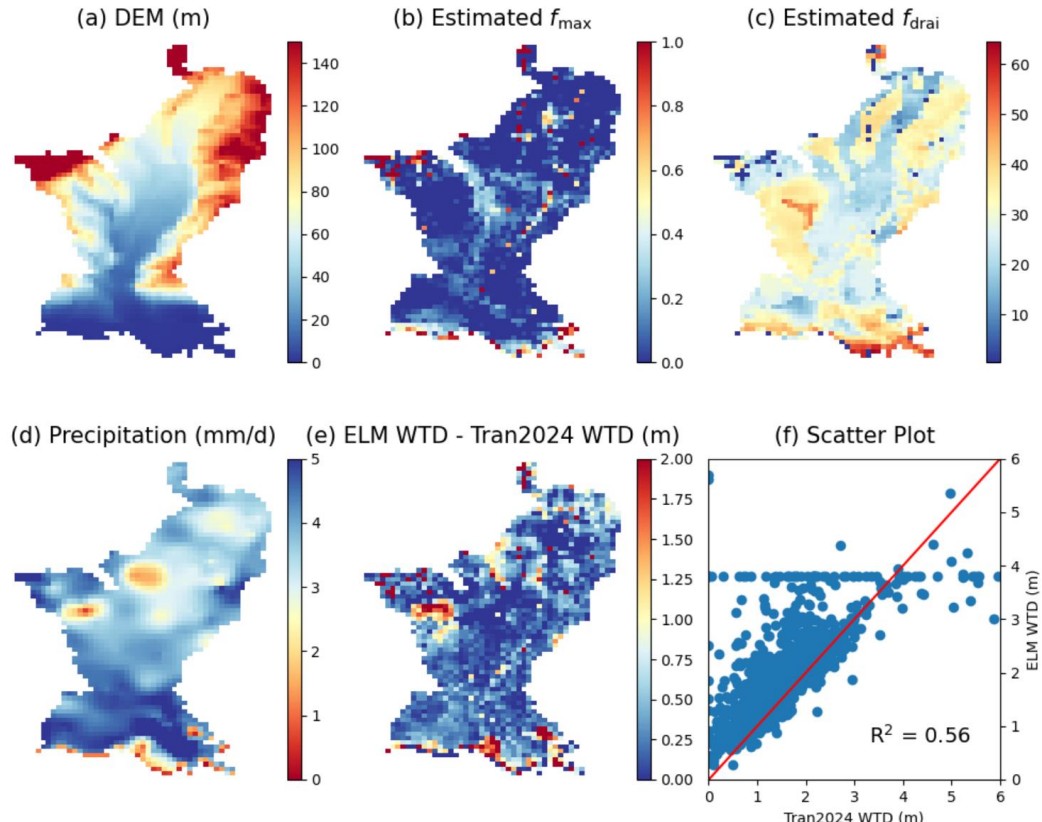

**Figure 3. Spatial distribution of surface elevation (a), $f_{max}$ (b), $f_{drai}$ (c), precipitation (d), and ELM**
**simulated water table depth compared to Tran2024 (e and f) in year 2015. Orange color in (d) represents less precipitation.**

### 3.2 Observation from the measurements

    To examine the impact of Hurricane Ida on the soil hydrology and nitrogen in the LMRB
region, we first analyze observations from measurements that are available within the region.





There are significant differences in the observed WTD at stations 302614091083001 and 302642091083401, although they are in close proximity to each other. However, Hurricane Ida influenced the observed water table depth at all selected monitoring stations (Fig. 4a), although with a weak signal. The river stage increased by more than 3 m after Ida made landfall at stations

07374000, 07380330, and 07380255 (Fig. 4b). The near shore stations 07381600 and 292800090060000 experienced less impact from Ida on the river stage. Water quality was also affected, with an increase in nitrogen concentration observed after Ida's landfall at both the inland station (07374000) and the nearshore station (07381600) (Fig. 4c). The diurnal variation of nitrogen concentration at station 07381600 disappeared during Ida, indicating a direct impact

from the elevated loss of nitrogen due to inland runoff. The rise in total N runoff at station 07374000 intensified following the landfall of Hurricane Ida (Fig. 4d), but gradually diminished as Ida progressed northeastward. The rise in total N runoff at 07381600 is weak. An increase in chlorophyll fluorescence was also observed during Ida near the estuary (Fig. 4e), which peaked at station 2928000090060000 shortly after Ida formed on August 26, and peaked at station

07380255 at Ida's landfall on August 29. Overall, observations revealed that Hurricane Ida reduced the water table depth and increased the river stage, the nitrogen concentration in the stream, the nitrogen runoff loading, and chlorophyll fluorescence in the estuary.



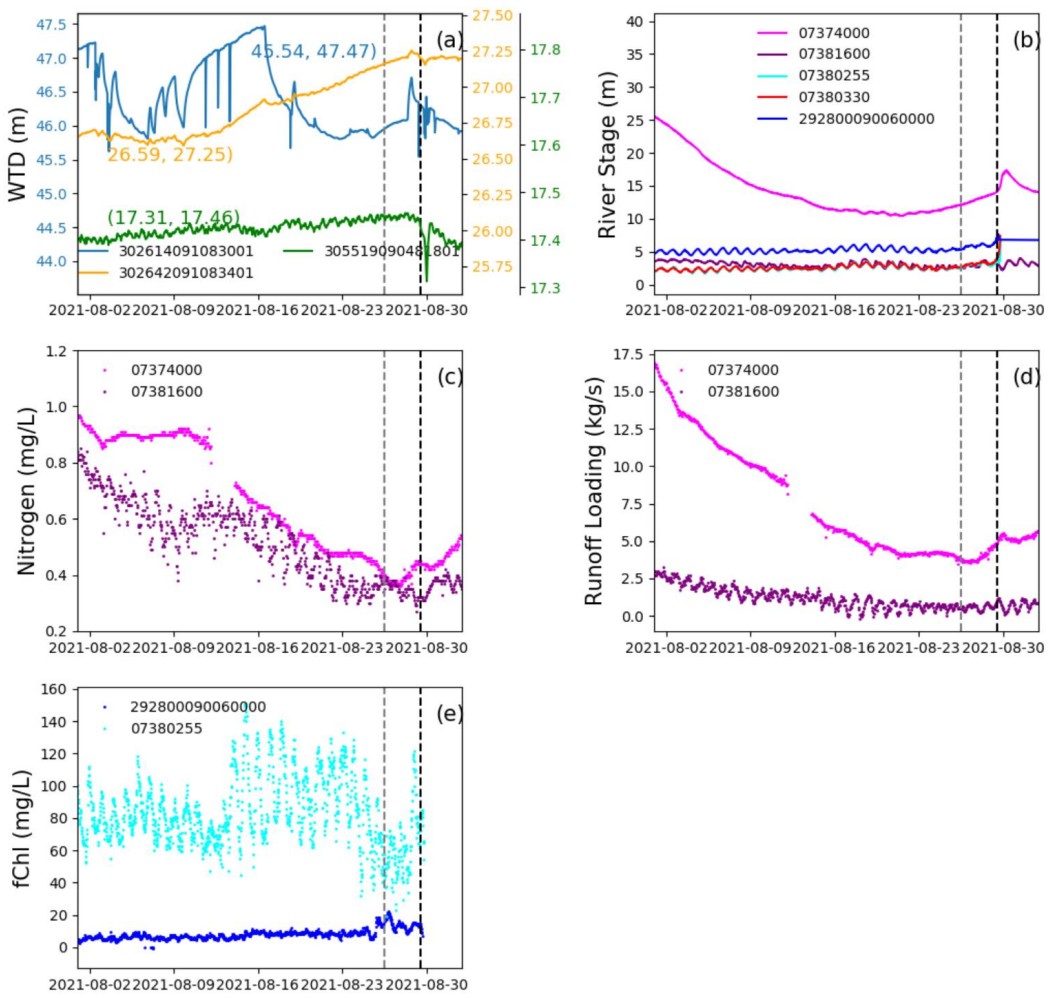

**Figure 4. Observed water table depth (a), river stage (b), nitrogen concentration (c), nitrogen runoff loading (d), and chlorophyll fluorescence (fChl) (e) at selected station locations shown in Figure 1. The numbers in the parentheses in (a) are the minimum and maximum values of WTD at each location in August 2021. The colors match station colors in Figure 1 for convenience. The three y-axes in (a) display the range of WTD at each station, with a corresponding legend color. The vertical grey dashed line represents the time Hurricane Ida formed. The vertical black dashed line represents the time Hurricane Ida made landfall.**

### 3.3 Impact of hurricane on modeled nitrogen loading in runoff

We examined water and nitrogen runoff loading related to Hurricane Ida simulated by ELM with the default and calibrated parameters ($f_{max}$ and $f_{drai}$) in the area affected by Ida. After Ida



formed, lower temperatures happened concurrently with abundant precipitation (Fig. 5a and Fig. 5b). The WTD from the calibrated model shows more pronounced response to changes in precipitation than the WTD from the model with default parameter values (Fig. 5c). By the end of August, after Hurricane Ida's landfall, WTD is reduced by 0.05 m and 0.31 m for the default

model and calibrated model, respectively (Fig. 5c). As crop irrigation from groundwater pumping is not considered in the modeling, the simulated WTD shown in Figure 5c is much shallower compared to the observations at the USGS stations (Fig. 4a) within the domain. Consistent with the rising water table, there is a notable increase in topsoil moisture evident in both models induced by the precipitation two days before and following the landfall of Hurricane

Ida (Fig. 5d).

During the whole period of August, the calibrated model has shallower water table and higher soil moisture compared to the default model, which results in higher total runoff consistently in August except after Ida's landfall (Fig. 5e). The two models noticeably respond differently to the heavy rainfall produced by Ida – the default model produced a larger increase

in soil moisture and higher runoff while the calibrated model produced a larger response in WTD and a more muted response in soil moisture and runoff. These differences are also reflected in a change in the surface runoff ratio before and after Ida's landfall (Fig. 5f). The calibrated model generally shows a higher surface runoff ratio than that of the default model in August until August 28 with the arrival of the first heavy rainfall event related to Ida, this ratio drastically

drops in the calibrated model while it remains about the same in the default model. Combining the changes in the total runoff and the surface runoff ratio suggest that the much smaller total runoff in the calibrated model compared to the default model shortly before and after Ida's landfall is mainly due to a much smaller surface runoff response to heavy rainfall in the calibrated model. Notably, the calibrated model has much smaller $f_{max}$ values compared to the

default model, which limit the surface runoff response to heavy rainfall while for smaller rain events, the calibrated model can still produce more surface runoff than the default model despite the smaller $f_{max}$ values due to its shallower groundwater table.

Differences in the soil hydrology response between the calibrated and default models may result in differences in the nitrogen response to Hurricane Ida. Corresponding to the increase in

soil moisture (Fig. 5d) and the total runoff generation (Fig. 5e) two days before Ida's landfall, the cumulative nitrogen (N) loss increased sharply on August 28, followed by smaller increases as




rainfall continued in the next few days (Fig. 6a). The larger increase in cumulative N loss in the default model is consistent with its higher total runoff compared to the calibrated model (Fig. 5e). Overall, nitrogen loss due to surface runoff constitutes a dominant portion of the total

nitrogen loss in both models (Fig. 6b) in majority of the time in August, especially between August 28 – 30 under the influence of rainfall associated with Ida. Although the surface runoff ratio drops significantly in the calibrated model after August 28, the fraction of surface N loss does not drop until after August 30, indicating a delayed N loss response relative to the runoff changes.

During low precipitation periods, nitrogen loss due to surface runoff constitutes a more dominant portion of total nitrogen leaching in the default model compared to the calibrated model (Fig. 6b), because of more concentrated surface soil mineral nitrogen due to drier soil in the default model. The total nitrogen leaching in August in the calibrated model is 80% of that in the default model, largely because the calibrated model has smaller N loss during Ida's heavy

rainfall events as limited by the smaller surface runoff response. Nitrogen leaching during Ida accounts for 38% and 31% of total nitrogen leaching in August for the default model and calibrated model, respectively.

Compared to the calibrated model, plant nitrogen uptake (Fig. 6c) and denitrification (occurs only in the anoxic fraction of soils) (Fig. 6d) in the default model are limited more by soil water,

as reflected by the drier soil in the default model (Fig. 5d), resulting in higher accumulation rate of nitrate in the soil (Fig. 6e), even though the default model simulates more runoff N loss (Fig. 6b). Denitrification declines more rapidly in the calibrated model than in the default model during Ida due to increased soil saturation. This results in a reduction of the anoxic fraction of soil, leading to a faster decrease in denitrification rates. As the cumulative nitrogen lost to runoff

exhibits a notable increase two days after Ida formed on August 26, the soil mineral nitrate-nitrogen drops sharply after August 26 (Fig. 6e).

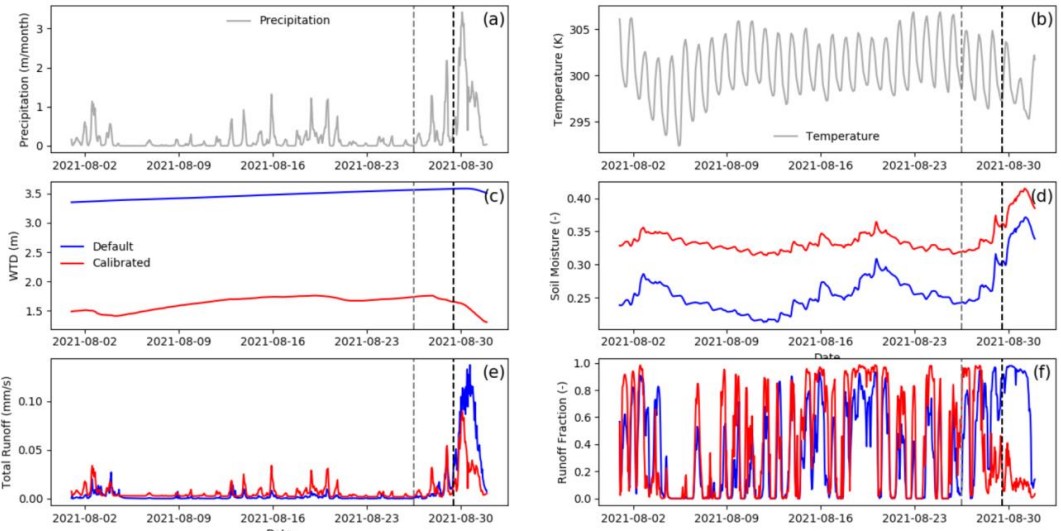

**Figure 5. Model inputs and comparison of simulations with default and calibrated parameters in August 2021 averaged within the Ida affected area, including: (a) precipitation, (b) air temperature, (c) water table depth (WTD), (d) average soil moisture of the topsoil layers, (e) total runoff, and (f) fraction of surface runoff in total runoff. Dashed grey line represents when Ida formed, and dashed black line is when Ida made landfall.**

The total nitrogen leaching (Fig. 6a) does not strongly correlate with precipitation (Fig. 5a) or runoff (Fig. 5e). There is a high leaching spike on August 24[th] in the calibrated model two days before Ida formed (Fig. 5e). On the previous day or August 23[rd], air temperature reached a maximum in August after a relatively dry period (Fig. 5b, 5d). Compared to the calibrated model prior to Ida formed, the dry stress (Figs. 5d and 6f) simulated by the default model caused a relatively faster increase in soil mineral nitrate-nitrogen (Fig. 6e) mainly due to lower plant nitrogen uptake and denitrification under stress (Fig. 6c).



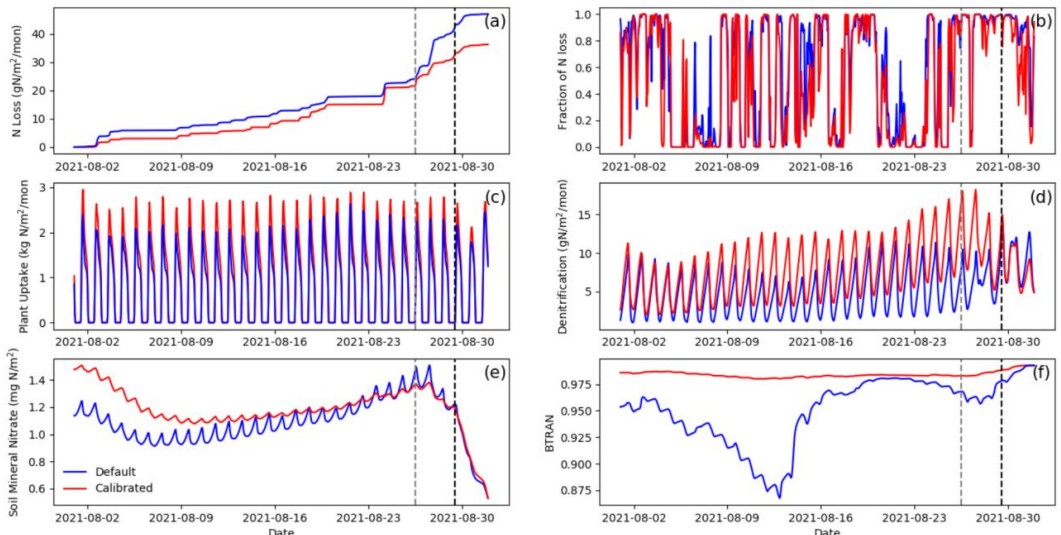

**Figure 6. Comparison between the default (blue) and calibrated (red) model results in August 2021 averaged within the Ida affected area: (a) cumulative sum of total nitrogen loss, (b) fraction of surface runoff, (c) total plant nitrogen uptake, (d) total denitrification flux, (e) soil mineral nitrogen, and (f) soil water stress factor, non-stressed when BTRAN =1. Dashed grey line represents when Ida formed, and dashed black line is when Ida made landfall.**

A notable increase in nitrogen leaching loss from the simulations was observed on August 24[th] (Fig. 6a). A spatial examination of the variables for the calibrated model on August 23[rd] and 24[th] revealed a shift of abundant precipitation towards the southeastern region near the Gulf Coast on the 24[th] (Figs. 7a, 7b). The increased runoff on the 24[th] (Fig. 7e), triggered by heavy precipitation, mobilized the accumulated nitrate (Fig. 7d) in the previously dry soil (Fig. 7c). This led to concentrated leaching in that area (Fig. 7f), explaining the spike in total nitrogen loss on August 24[th] shown in Figure 6a. Although this event occurred before hurricane Ida, these findings underscore the significance of considering preceding environmental conditions in understanding the hurricane impact on nitrogen leaching loss.

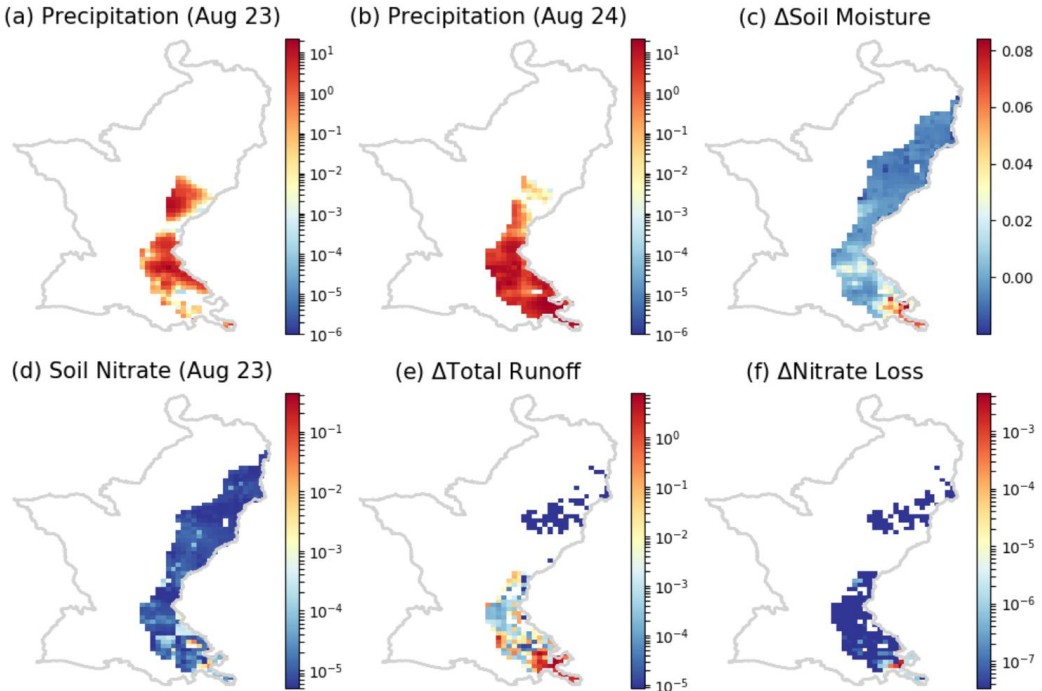

**Figure 7. Comparison of variables between August 23rd and 24th for the calibrated model:**
**(a) precipitation (mm/d) on August 23rd, (b) precipitation (mm/d) on August 24th, positive**
**increase of soil moisture (m³/m³)(c), soil nitrate-nitrogen (gN/m²) on August 23rd (d),**
**positive increase in runoff (mm/d) (e), and positive increase in nitrate-nitrogen leaching**
**loss (gN/m²/d) (f) from August 23rd.**

To understand the driving mechanism of nitrogen leaching loss under different soil water
conditions, we selected a nearshore grid in the subdomain affected by Ida from the two
simulations which have different level of soil water as an example. At this selected grid, unlike
the conditions averaged over the Ida affected area discussed in Fig. 5 and Fig. 6, the calibrated
model happens to have drier soil (Fig. 8b) compared to the default model (Fig. 8a). As a result of
the smaller rate of denitrification and plant uptake as soil nitrate-nitrogen sink and the reduced
source from nitrogen fixation due to water stress, there is more soil nitrate accumulation in the
calibrated model (Fig. 8d). Soil water not only affects the competition for nitrogen between the
plant and soil microbes, but also the vertical transport of soil mineral nitrogen. Compared to the
default model, drier soil from the calibrated model due to larger $f_{max}$ at this grid favors higher
surface runoff before Hurricane Ida formed. Drier soil also causes slightly warmer temperature.
It is not shown because the contrast is not visually discernible. This warmer temperature can



partially offset the deceleration of soil organic matter decomposition caused by dry soil conditions. After Ida induced precipitation in the area, infiltration pushed the accumulated nitrogen further down to the deeper soil layers (Fig. 8d). More nitrogen from the calibrated model leached through the pathway of subsurface runoff (Fig. 8f) during Ida even though the subsurface runoff is far less than that from the default model.

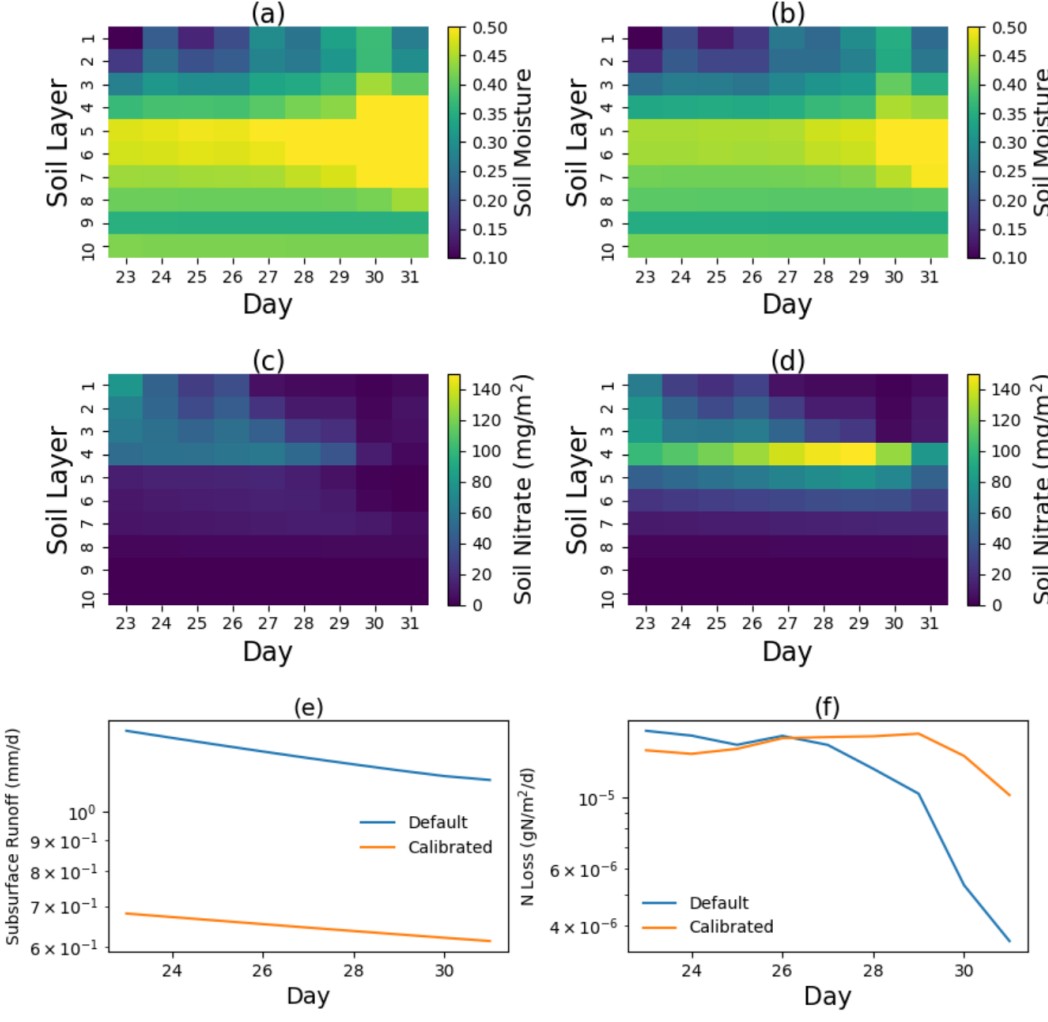

**Figure 8. Heatmap comparison of soil moisture, soil temperature, and soil mineral nitrate-nitrogen between the default model (a,c,e), the calibrated model (b,d,f), and nitrogen leaching loss (g) at a selected nearshore grid.**





## 4. Discussions

### 4.1 Importance of soil hydrology on nitrogen leaching

By conducting two simulations with the ELM model using default and calibrated parameters
that influence surface and subsurface runoff processes, our results revealed that soil hydrology
can have a large impact on nitrate-nitrogen riverine loading through surface and subsurface
runoff when there is a significant concentration of nitrogen in soil water and sufficient recharge
through rainfall or irrigation (Meisinger and Delgado, 2002). The dynamics of nitrogen riverine
loading is linked to the movement of water through the soil profile. Adequate soil moisture levels
promote microbial decomposition of organic matter and subsequent release of nitrogen into the
soil. Different parameterizations of subsurface and surface runoff can have a significant impact
on the nitrogen dynamics in the soil and consequently loss through the runoff.

The interplay between soil moisture dynamics and nitrogen transport is critical for
understanding riverine nitrogen loading, especially in the context of short-term impacts from
hurricane events when soil moisture can experience high-frequency variability. When surface
runoff dominates, nitrogen can be rapidly transported to surface water, leading to spikes in the
riverine. On the other hand, nitrogen is slowly leached through the soil profile and into
groundwater systems before eventually reaching rivers, showing a delayed response compared to
the surface runoff loss. Both model simulations in this study suggest dominant nitrogen loss
through surface runoff, and the response to hurricane in WTD and nitrogen runoff loading is
approximately consistent with the observations. This suggests ELM can provide some valuable
insights into the mechanisms driving nitrogen runoff loading during hurricanes.

While the focus of this study is primarily on analyzing the response of nitrogen runoff
loading during hurricane Ida using ELM, it is also important to consider the pre-existing
conditions of soil water for water quality management. This is particularly crucial for nearshore
areas which are dry prior to hurricane landfall, as the potential harm to water quality can be
particularly acute. By understanding both aspects (prior and during the hurricane), we can better
anticipate and mitigate the adverse effects of hurricane in the vulnerable regions.

### 4.2 Model limitations

Although the models successfully captured the groundwater table response to Hurricane Ida's
landfall, consistent with observations from the monitoring wells that show a rising water table,



our findings revealed faster diminishing of nitrogen runoff after Ida's landfall on August 29[th] in the area affected by Hurricane Ida. The model indicated significantly higher nitrogen leaching before Ida's landfall, earlier than the water quality observations at monitoring stations. The early

nitrogen runoff was attributed to the abundant precipitation in southern LMRB in late August 2021 before and after Hurricane Ida formed on August 26[th], leading to elevated soil moisture levels prior to Ida's landfall. This precipitation mobilized nitrogen in the previously dry and warm soil, as indicated by the model results, leaving less soil mineral for leaching after Ida's landfall.

Apart from the heterogeneity in precipitation both in time and space, the discrepancy in WTD and nitrogen leaching between the simulations and observations largely stem from the omission of crop management factors in the model. As one of the largest agricultural crop-producing areas in the U.S. (Tiwari *et al.*, 2023), the LMRB experiences significant agricultural activities that contribute to nitrogen dynamics. Over 90% of the groundwater used for irrigation

in the LMRB comes from the Mississippi River Valley Alluvial Aquifer (Reba and Massey, 2020). The absence of accounting for these agricultural practices could be a factor influencing the observation-model inconsistencies in nitrogen leaching patterns, as evidenced from the results between our two model simulations with distinct WTDs.

    Another important overlooked process in the model is the lateral transport of nitrogen

by advection and diffusion induced by hydrologic connectivity and strong nitrogen gradient between neighboring grid cells. The timing of hydrologic connectivity and nutrient gradients may affect a range of downslope nutrient transport and biogeochemical transformation along the topo sequence (Stieglitz *et al.*, 2003; Kelly *et al.*, 2021).

## 5. Conclusions

In conclusion, we calibrated two parameters associated with surface and subsurface runoff in the ELM using the water table depth (WTD) obtained from a previous 3D hydrology simulation in the Lower Mississippi River Basin (LMRB). We then compared the nitrogen runoff leaching results from the calibrated ELM model with those from the default ELM model. Our analysis of WTD in the calibrated model and the 3D model revealed that despite model calibration to match

the ELM WTD with that simulated by the 3D model, neglecting lateral flow in ELM can still result in noticeable difference between the WTD in the two models, particularly in slope areas



with limited precipitation. The calibrated ELM was able to simulate the increased nitrate-
nitrogen runoff leaching during Hurricane Ida, as evidenced by water quality and hydrologic
observations within the affected region. However, the timing of peak leaching and the leaching
pathways can be influenced by factors such as soil moisture, soil temperature, precipitation, and
lateral transport. Thus differences between the calibrated and default models as well as
differences between the models and observations (e.g., WTD, crop management) can result in
differences between the observed and simulated nitrogen response to Hurricane Ida. Even though
the model captures the N runoff loading signal in the affected area by Hurricane Ida, the current
lack of lateral transport of nitrogen within the soil and in the river in ELM hinders the realistic
prediction of nitrogen runoff loading in response to hurricanes.

## Cod and Data availability

The code is available at https://zenodo.org/uploads/11372002 (Fang, 2024a). The neural network
model script, data used to train the model, and the observation data used in this study are
available at https://doi.org/10.5281/zenodo.10927512 (Fang , 2024b).

## Author contributions

YF designed, conducted the experiments and analysis, and drafted the manuscript. HT provided
the water table depth data from the ELM-ParFlow simulation, contributed to machine learning,
and manuscript editing. LRL contributed to result interpretation and discussion, as well as
manuscript editing.

## Competing interests

The contact author has declared that none of the authors has any competing interests.

## Acknowledgments

This study was supported by the U.S. Department of Energy Advanced Scientific Computing
Research (ASCR) program through the Multiphysics Simulations and Knowledge discovery
through AI/ML technologies (MuSiKAL) project. The simulations presented in this article were
performed on computational resources managed and supported by the Department of Energy



(DOE) Office of Biological and Environmental Research (BER) and housed in PNNL's
Computational Science Facility. PNNL is operated for DOE by Battelle Memorial Institute under
Contract DE-AC05-76RL01830.

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
