# Peer review of "Short-term effects of hurricanes on nitrate-nitrogen runoff loading: a case study of Hurricane Ida using E3SM land model (v2.1)"

_Geoscientific Model Development, 2024_

## Author Comment (AC1)

Dear Editor and Reviewers,

We appreciate your time and effort handling and reviewing our manuscript. Thank you for your thoughtful comments, which have been very helpful in improving the manuscript. In the following, we have addressed each comment of the two reviewers in blue text. The text added in the revision is shown in italics in this response.

**Response to RC1**

The manuscript proposed new hydrological parameterization for ELM in the Lower Mississippi river basin by constraining ELM predicted water table depth with predictions from a calibrated 3D subsurface hydrology model. Then the manuscript conducted simulations during the period of Hurricane Ida using both default and new parameterization with ELM and compared the simulated hydrological and nitrogen responses to Ida with observations. The study concludes runoff process is important to predict hydrological and nutrient responses to heavy rainfall events.

Overall, the paper is easy to read. However, One thing is unclear to me - what is the key evidence that calibrated ELM is better than default ELM? It seems to me the biggest difference is that default model does not predict changes in WTD but calibrated model predicted shallower WTD after Ida. The conclusion states that "the calibrated ELM was able to simulate the increased nitrate-nitrogen runoff leaching during Hurricane Ida" (Line 472-473) but both versions predict increases in runoff and N Loss (Fig. 5e and Fig. 6a) and it is not discussed which one is closer to the observation. Given this is a model evaluation paper, more direct and quantitative model-data comparison would be helpful

**Thanks, we are glad the reviewer found our manuscript easy to read. Below are our point-by- point responses to your comments.**

One thing is unclear to me - what is the key evidence that calibrated ELM is better than default ELM? It seems to me the biggest difference is that default model does not predict changes in WTD but calibrated model predicted shallower WTD after Ida.

We apologize for the oversight in not including the uncalibrated results. As can be seen from the figure below, the uncalibrated model resulted in WTD ranging between 3 and 5 m and performs poorly in simulating the spatial distribution of WTD when compared to Tran2024.

We added this statement in Section 3.1 in the revision for clarification:

When running the uncalibrated ELM, the simulated WTD ranged between 3 and 5 m, with an  $R^2$  value of -1.66 when compared to Tran2024, indicating poor performance in simulating the spatial variations of WTD.

Some additional comments:

Line 195-205. The whole training workflow reads a little confusing to me. WTD is a model ouptut depending on fmax and fdrai but it is used to predict fmax and fdrai... Doesn't it make more sense to create a model/emulator for WTD based on ELM inputs and parameters and then find the best fmax/fdrai to get WTD values close to Tran2024 WTD?

We agree that running multiple ELM simulations with varied  $f_{max}$  and  $f_{drai}$  parameter values to train an emulator which can then be used to find the optimal value could be an effective method for closely matching the observed WTD values from Tran2024. To produce training data for the emulator requires a large ensemble of simulations with perturbed parameters, which can be computationally expensive. Instead, we opted for the iterative approach described in Section 2.4. This method allowed us to efficiently calibrate the parameters. We have added the following justification to the revised manuscript.

The parameter estimation can be achieved by running multiple ELM simulations with varied  $f_{max}$  and  $f_{drai}$  parameter values to train an emulator that can be used to find the optimal parameter

values. While this could be an effective method for closely matching the observed WTD values from Tran2024, it needs a substantial number of simulations to produce the training data, which can be computationally expensive. Instead, we explored an alternative approach by using machine learning.

Line 285. For nitrogen and runoff loading response, it seems only one station has significant response but the other one (07381600) did not? Any explanations? In addition, is the model able to explain the decline of Nitrogen and Runoff Loading before Ida?

Thanks for this comment. As described in line 170 in the original submission, station 07381600 is a near shore station, which has more influence from the ocean as can be seen from the river stage in Figure 2b.

Unfortunately, the model cannot be used to explain the decline of Nitrogen and Runoff loading before Ida. The primary focus of our study was to investigate the hurricane's effect on runoff nitrogen loading. Simulating the entire dynamics of nitrogen within the stream requires in-stream N transport and obtaining data on detailed N utilization in the area, which is beyond the scope of this study. To address the comment, we acknowledged this limitation in the revised manuscript at the end of section 4.2 and highlighted that future model developments could be used to explain such dynamics.

Note that the current version of the ELM does not include an in-stream nitrogen model, which is under active development. Consequently, while our study successfully examines the impact of hurricanes on runoff nitrogen loading, it does not fully capture the complete dynamics of nitrogen in stream before and after the hurricane, and hence it cannot be used to explain the observed N loading dynamics before Ida. Future improvements to the ELM, including the integration of an in-stream N model, will enable more detailed and accurate simulations of nitrogen and runoff loading dynamics in rivers.

Line 370. Fig. 6 panel a, if the value is cumulative sum, the unit should just be gN/m2 without per month.

Thanks for catching the error. We removed "per month" in the figure.

Line 410. There is not panel g in Fig. 8

Thanks for the careful review! We have corrected to caption as

Figure 8. Heatmap comparison of soil moisture and soil mineral nitrate-nitrogen between the default model (a,c), the calibrated model (b,d), subsurface runoff (e) and nitrogen leaching loss (f) at a selected nearshore grid.

Line 432-433. What are the "valuable insights"?

We have revised this sentence for clarity:

This suggests ELM can provide some valuable insights into the mechanisms driving nitrogen runoff loading during hurricanes, such as the significant role of surface runoff and changes in water table depth.

**Response to RC2**

The Manuscript developes a new soil hydrology parameterization for ELM for the Lower Mississipi River Basin (LMRB) in an effort to better understand nitrate loss. This proposed calibrated model predicts water table depth from a 3D subsurface hydrology model. The authors then conducted simulations of the Hurricane Ida of period and compares default ELM and calibrated model simulated results with observations.

I think the manuscript is relatively easy to read and understand and proposes a novel way to improve our understanding of nitrate leaching. However, I struggle to understand which model performs better in the context of the observations. Both versions predict an increase nitrate runoff but having these comparisons with direct observations on the figure would help the reader understand how model performance compared.

We thank you for your kind and encouraging words and your recognition of our novel approach to improving the understanding of nitrate leaching.

The primary focus of our study was to investigate the hurricane's effect on runoff nitrogen loading. Unfortunately, we were not able to use the model for direct comparison with observations as the model cannot simulate in-stream N transport. We acknowledged this limitation in the revised manuscript at the end of section 4.2 and highlighted that future model developments could be used to explain such dynamics.

Note that the current version of the ELM does not include an in-stream nitrogen model, which is under active development. Consequently, while our study successfully examines the impact of hurricanes on runoff nitrogen loading, it does not fully capture the complete dynamics of nitrogen in stream before and after the hurricane, and hence it cannot be used to explain the observed N loading dynamics before Ida. Future improvements to the ELM, including the integration of an in-stream N model, will enable more detailed and accurate simulations of nitrogen and runoff loading dynamics in rivers.

**Additional comments**

Figure 1 make the USGS points larger to improve clarity

**Increased the size of the USGS points. Thanks!**

In figure 3f, I understand the overprediction but why are there a bunch of points at just less than 4m?

Thanks for the observation and question. These are probably points that cannot obtain an optimal solution simply by tuning two model parameters. Comparing the WTD of figure 5C to observations in 4A was difficult and model results and observations seem very different.

Thanks for the comment. We didn't intend to use the model to compare with observations as it requires additional data and model development, which is beyond the scope of this study. We added this sentence in the Introduction for clarification:

We will not delve into the full dynamics of nitrogen cycling within the stream due to the limitation of the model, as addressing the limitation is beyond the scope of this study.

line 232 Specify what exactly those insight are

Thanks for the comment! We assumed you meant line 432 in the original submission. We added further details for clarification in the revised manuscript, which is repeated below:

This suggests ELM can provide some valuable insights into the mechanisms driving nitrogen runoff loading during hurricanes, such as the significant role of surface runoff and changes in water table depth.

Line 382 Cod"e"

Thanks! Corrected to "Code".

The biggest issue with the paper was understanding whether the default model or the calibrated model performed better.

The calibrated model performs better in WTD as can be seen from the figures for the default model (left panel) and calibrated model (right panel). The model itself was not used to quantitatively compare the N loading from model simulations with the observations for the reason provided in our response to your main comments.

---

## Author Response (AR2)

Dear Editor and Reviewer,

We thank you again for your insightful comments, which have helped us to improve our manuscript. Below please find our point-by-point response to the reviewer's comments, shown in blue.

**Response to Reviewer's Comments**

The revision has improved the clarification of the study. I have two more comments about the framing of the study.

Thank you for taking the time to review our revised manuscript and provide us with your thoughtful comments about the framing of our study. We are glad that you found our previous revision improved the clarity of our study.

1. My key take-away of the study is that subsurface parameterization is important to simulate hurricane response and the study proposes a new iterative method to improve the parameterization. However, this is not evident at all in the title, which only states nitrogen loading. So the title sounds like some biogeochemical improvement in modeling… I would suggest to change the title to reflect the importance of subsurface hydrological parameterization.

We appreciate your suggestion to revise the title to better reflect the importance of subsurface hydrological parameterization in simulating hurricane response. Here is our revised title:

Subsurface hydrological controls on the short-term effects of hurricanes on nitrate-nitrogen runoff loading: a case study of Hurricane Ida using E3SM land model (v2.1)

2. It would be helpful to include some more in-depth discussion on whether the iterative approach (key novelty in my understanding) can be extended to other watersheds.

Thanks for this great suggestion! We have added a subsection in the Discussion section to address this point. In it, we discussed the prospects and challenges of applying the iterative approach to other watersheds. We repeated it here:

4.1 Potential applications of the iterative parameterization approach

The iterative parameterization approach presented in this study demonstrates a promising method for improving subsurface hydrological simulations and can be easily extended to other watersheds. The use of a surrogate model to estimate model parameters reduces computational

costs while maintaining accuracy, allowing for efficient iterative refinement of the simulation results. This approach can be particularly beneficial for watersheds with complex hydrogeological characteristics, where traditional calibration methods may be computationally prohibitive or require extensive data sets.

Successful application of this method relies on prior knowledge of the most sensitive and important parameters to include in the parameterization process. Without a clear understanding of which parameters have the greatest impact on model predictions, the iterative approach may not effectively reduce uncertainty or improve simulation accuracy. Additionally, identifying key parameters can help to avoid over-parameterization, where the inclusion of too many parameters can lead to overfitting and degradation of predictive performance.